# Pharmacokinetics of hydrogen administered intraperitoneally as hydrogen-rich saline and its effect on ischemic neuronal cell death in the brain in gerbils

**Momoko Hirano[1], Kazuhisa Sugai[1], Masahiko Fujisawa[1], Eiji Kobayashi[2,3,4], Yoshinori Katsumata[2,3], Yoji Hakamata[1,2]\*, Motoaki Sano[2,3]\***

**1** Department of Basic Sciences, School of Veterinary Nursing and Technology, Faculty of Veterinary Sciences, Nippon Veterinary and Life Science University, Tokyo, Japan, **2** Center for Molecular Hydrogen Medicine, Keio University, Tokyo, Japan, **3** Department of Cardiology, Keio University School of Medicine, Tokyo, Japan, **4** Department of Organ Fabrication, Keio University School of Medicine, Tokyo, Japan

\* msano@a8.keio.jp (MS); yhakama@nvlu.ac.jp (YH)

## Abstract

Intraperitoneal administration of hydrogen ($H_2$)-containing saline inhibited neuronal cell death in ischemic stroke in a number of animal models, but it is unknown whether $H_2$ is absorbed from the abdominal cavity into the blood and reaches the brain. In this study, we investigated whether intraperitoneal administration of saline containing $H_2$ inhibits neuronal cell death caused by cerebral ischemia and measured the concentration of $H_2$ in the carotid artery and inferior vena cava (IVC). Gerbils were subjected to transient unilateral cerebral ischemia twice, and saline or $H_2$-rich saline was administered intraperitoneally three or seven times every 12 hours. We evaluated the number of apoptotic cells in the hippocampus and cerebral cortex on day 3 and the number of viable neurons in the hippocampus and cerebral cortex on day 7. In addition, a single dose of saline or $H_2$-rich saline was administered intraperitoneally, and blood $H_2$ levels in the carotid artery and IVC were measured. On day 3 of ischemia/reperfusion, the number of neurons undergoing apoptosis in the cortex was significantly lower in the $H_2$-rich saline group than in the saline group, and on day 7, the number of viable neurons in the hippocampus and cerebral cortex was significantly higher in the $H_2$-rich saline group. Intraperitoneal administration of $H_2$-rich saline resulted in large increases in $H_2$ concentration in the IVC ranging from 0.00183 mg/L (0.114%) to 0.00725 mg/L (0.453%). In contrast, carotid $H_2$ concentrations remained in the range of 0.00008 mg/L (0.0049%) to 0.00023 (0.0146%). On average, $H_2$ concentrations in carotid artery were 0.04 times lower than in IVC. These results indicate that intraperitoneal administration of $H_2$-rich saline significantly suppresses neuronal cell death after cerebral ischemia, even though $H_2$ hardly reaches the brain.

**Data Availability Statement:** All relevant data are within the article and its Supporting Information files.

**Funding:** This work was supported by grants from Doctors Man Co., Ltd. (M.S.). Sou Hashimoto (Doctors Man Co., Ltd.) provided us with the H2 filling device. The funders had no role in study design, data collection and analysis, decision to publish, or preparation of the manuscript.

**Competing interests:** MS and EK received advisory fees from Doctors Man Co., Ltd. Furthermore, MS and EK are the registered inventors of the patent jointly filed by Keio University and Doctors Man, "Methods for generating organ preservation solution containing hydrogen and organ preservation solution containing hydrogen" (Application number PCT/JP2019/045790). This does not alter our adherence to PLOS ONE policies on sharing data and materials.

# Introduction

Molecular hydrogen ($H_2$) is the most abundant molecule in the universe, but its medical value has long gone unnoticed. Its effectiveness in medicine was first studied in a rat cerebral infarction model [1]. Since then, its efficacy has been demonstrated in many pathologies, especially those involving ischemia-reperfusion injury [2–9].

For a long time, the effect of hydrogen in reducing ischemia-reperfusion injury was thought to be based on scavenging of reactive oxygen species [1]. However, other mechanisms also may be involved [7, 10].

Although the efficacy of molecular $H_2$ has been reported extensively, its pharmacokinetics and pharmacodynamics after administration were first investigated only recently: We performed a study in which we measured the blood concentration of $H_2$ in miniature pigs after inhalation of $H_2$ [5, 11] and injection of $H_2$ water into the intestinal tract [4]; the study showed that after a single inhalation of $H_2$ gas, the $H_2$ concentration in carotid arterial blood rises quickly but drops below the detection limit within a few minutes [12]. $H_2$ in the blood appears to be rapidly excreted through exhalation. By administering 500 ml of $H_2$-rich water through a tube inserted into the jejunum of miniature pigs, we confirmed that $H_2$ was absorbed into the blood through the jejunal wall and flowed into the portal vein and hepatic superior inferior vena cava (IVC); however, almost all of it was expired and only a little reached the carotid artery [4].

To date, studies examining changes in the blood concentration of $H_2$ have been conducted only in pigs. Furthermore, to the best of our knowledge, no studies have examined this topic in rodents. However, rodent experiments would be important because these animals are frequently used in basic medical research, including as disease models.

Rodents are often used for basic experiments on cerebral infarction, but the anatomy of their cerebral collateral blood vessels varies, e.g., at the circle of Willis; these differences are important because collateral hemodynamics have a significant influence on experimental generation of cerebral ischemia. Gerbils do not have an anastomosis between the carotid and vertebrobasilar arteries at the circle of Willis, so unilateral common carotid occlusion in the neck can easily create ipsilateral cerebral ischemia (**S1 Fig**) and they are often used in ischemic stroke research [13]. Therefore, previously we established a model of cerebral ischemia in gerbils in which the only required manipulation is transient occlusion of the unilateral common carotid artery in the neck; consequently, this model has low variability of data and enables reliable testing of drug efficacy [14–16].

In this study, $H_2$-rich saline solution was administered intraperitoneally to gerbils to evaluate whether $H_2$ inhibits ischemic neuronal cell death and to assess the pharmacokinetics of $H_2$.

# Materials and methods

## 2.1 Ethics statement

This study was approved by the Institutional Animal Care and Use Committee in Nippon Veterinary and Life Science University (No. 2021K-19). All animal experiments complied with the ARRIVE guidelines and were performed in accordance with the National Research Council's Guide for the Care and Use of Laboratory Animals.

## 2.2 Animals and experimental group

Male (n = 54) and female (n = 54) Mongolian gerbils (body weight [BW], 70–90 g) were used. All animals were bred in our laboratory, maintained in pathogen-free environments, and provided with food and water ad libitum. The carotid artery was clipped in 90 of the 108 gerbils, and 22 ischemia-positive animals were subsequently selected (see below). No cerebral ischemia

was induced in the remaining gerbils (n = 17), and they were used for staining normal brain tissue and measuring blood $H_2$ concentration.

No criteria were set for including or excluding animals, and no potential confounders were identified. The researchers were not blinded to group allocation, but the experiments and data analysis were conducted by different people to avoid potential bias.

## 2.3 Preparation of $H_2$-rich saline

To prepare $H_2$-rich saline, a polyethylene terephthalate bottle was filled with saline; then, $H_2$ was pressurized with a DAYS hydrogen gas filling system (Doctors Man Co., Ltd., Japan) and filled into the bottle to a gauge pressure of 0.4 MPa, as described previously [4]. The bottle was shaken for 30 seconds to dissolve the $H_2$, and then the lid was opened to reduce the pressure to atmospheric pressure. $H_2$-rich saline ($H_2$ concentration, 2.6 mg/L) was prepared immediately before each set of injections.

## 2.4 Cerebral ischemia model in gerbils and experimental protocol

Gerbils were anesthetized with 2% isoflurane, and the left carotid artery was exposed by a mid-neck incision and occluded with a vascular clip. Anesthesia was immediately stopped, and the animals were awakened. Each animal's behavior was carefully observed in the open field for 10 minutes, during which time neurological signs and the stroke index [17] were recorded. Animals with a stroke index of 10 or greater (scale, 0–25) were classified as ischemia positive and used in experiments to test the effects of intraperitoneal administration of $H_2$-rich saline. The ischemia-negative animals were not used in subsequent experiments. In the ischemia-positive animals, the clip was released after 10 minutes of occlusion to restore blood flow, and the occlusion was repeated after 5 hours. After the second occlusion, the animals were randomly divided into $H_2$-rich and saline groups. Animals in each group received repeated intraperitoneal injection with 20 ml/kg BW of solution three or seven times at 12-hour intervals. We examined two administration frequencies with the aim of confirming the existence of a dose-response relationship (**S2 Fig**).

## 2.5 Assessment of cerebral neurons

To assess cerebral neurons, 15 of the ischemia-positive animals were selected and assigned to the $H_2$-rich saline group (n = 7) or the saline group (n = 8).

At day 7 after ischemia-reperfusion, the animals were anesthetized by intraperitoneal injection of pentobarbital (Kyoritsu Seiyaku, Japan, 65 mg/kg BW). The transcardiac approach was used to perfuse brains with heparinized saline followed by 10% phosphate-buffered formalin. Then, brains were further fixed overnight in 10% phosphate-buffered formalin, embedded in paraffin and cut into 5-μm sections and stained with hematoxylin and eosin (HE). Each brain section was scanned with a NanoZoomer-SQ (Hamamatsu Photonics K.K., Japan) x 40 mode (0.23 μm/pixel), and images were analyzed by NDP view software (NDPver2.0, Hamamastu Photonix K.K., Japan) to measure the number of neurons in the hippocampal CA1 area and layer III and V of the cerebral cortex in each hemisphere (**S3 Fig**). The area of measurement in each hemisphere was defined as a 300-μm wide column between the interhemispheric and rhinal fissures, as described previously [14].

## 2.6 Terminal deoxynucleotidyl transferase dUTP nick end labeling apoptosis assay

Apoptosis is involved in neuronal death after ischemia-reperfusion injury [18, 19], so we investigated whether it occurred in our gerbil model of cerebral ischemia-reperfusion injury and whether it is inhibited by intraperitoneal administration of $H_2$-rich saline.

To assess apoptosis, 7 of the ischemia-positive animals were selected and randomly assigned to the $H_2$-rich saline group (n = 4) or the saline group (n = 3).

On day 3 after ischemia-reperfusion, animals were anesthetized by intraperitoneal injection of pentobarbital (65 mg/kg BW; Kyoritsu Siyaku, Japan,). Brain sections were prepared as described above. Terminal deoxynucleotidyl transferase dUTP nick end labeling (TUNEL) staining was performed on paraffin-embedded sections with an ApopTag® *in situ* apoptosis detection kit (Sigma-Aldrich Co., USA) according to the manufacturer's protocol. After staining, counterstaining of nuclei was performed with hematoxylin. The method of Hanyu et al. [14] was used to image a 300-μm wide area midway between the longitudinal cerebral column and the left and right rhinal fissure in cerebral cortex and a 300-μm wide area in the hippocampus with NanoZoomer-SQ. The number of TUNEL-positive cells in layer III and V of the cerebral cortex and hippocampus was measured with image processing software (NDPver2.0, Hamamastu Photonix K.K., Japan).

## 2.7 Measurement of blood $H_2$ concentration

To determine whether the hydrogen in $H_2$-rich saline reaches the brain after intraperitoneal injection, blood samples were taken from the IVC and common carotid artery, the main vessels connecting the injection site (intraperitoneal) and the injured brain. Gerbils were anesthetized with isoflurane and injected intraperitoneally with 20 mL/kg BW of $H_2$-rich saline (n = 9) or saline (n = 6). After 5 minutes, blood was collected from the carotid artery, and after 10 minutes, from the IVC. The blood $H_2$ concentration was measured as described previously [4]. Briefly, to measure blood $H_2$ concentration, a needle was first inserted into the rubber lid of a 13.5-mL sealed vial, 1 mL of air was removed, and 1 mL of blood was injected. The rubber lid was immediately coated with wax to seal the injection hole. In such a sealed vial, $H_2$ in the blood is released into the air phase; therefore, the $H_2$ concentration in the blood was estimated by measuring the $H_2$ gas concentration in the air in the vial with a gas chromatograph (TRIlyzer mBA-3000, Taiyo Co.). Calibration curves were prepared with $H_2$ gas concentrations of 0 (nitrogen gas), 5, 50, and 130 ppm. Samples were taken from the carotid artery and IVC, and each sample was measured twice. At the same time, the $H_2$ gas concentration was measured in the air in vials in which no blood was injected. The $H_2$ concentration in the air was 0.5 ppm, i.e., above the quantification limit. Therefore, the value in the air was subtracted from each sample measurement.

## 2.8 Statistical analysis

The number of cerebral neurons is presented as the ratio of neurons in the left and right hemispheres. Other data are presented as mean ± standard error. Experimental groups were compared by one-way analysis of variance (ANOVA) followed by the Tukey-Kramer multiple comparison test. *P* values below 0.05 were considered as significant.

## Results

All ischemia-positive animals were eligible for inclusion in the analyses.

### 3.1 Number of surviving neurons in hippocampal tissue after ischemia-reperfusion

Ischemia-reperfusion injury markedly reduced the number of surviving neurons in the hippocampal CA1 region compared with the contralateral healthy region (**Fig 1A and 1B**). Surviving neurons had large round nuclei and well-defined nucleoli. Dead neurons had shrunken nuclei

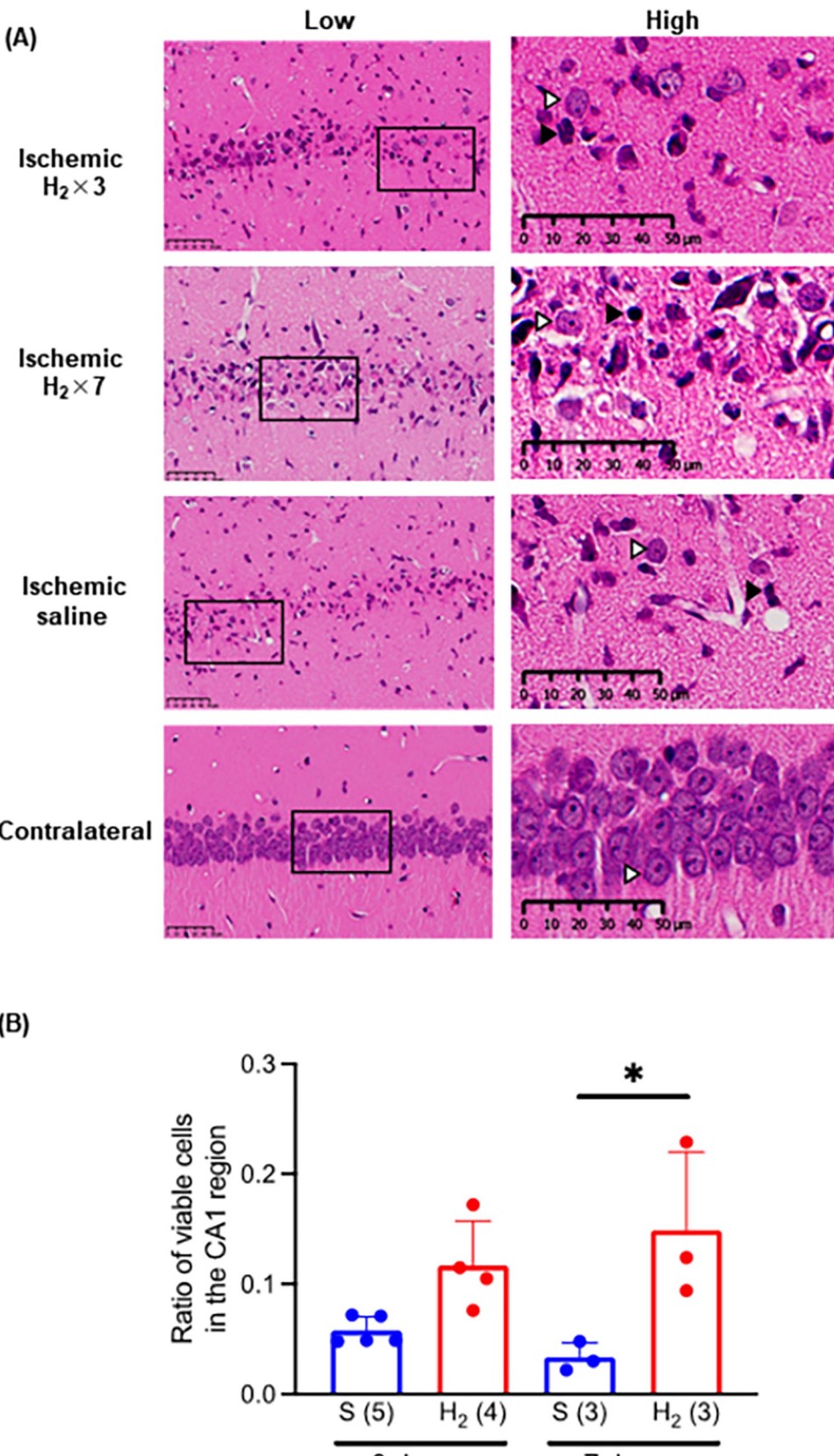

**Fig 1. Effect of intraperitoneal injection of $H_2$-rich saline on ischemic injury in hippocampal region CA1 in gerbils.** (A) Hematoxylin and eosin staining of ischemic CA1 and contralateral (healthy) regions. The image on the right is an enlarged view of the inset in the image on the left. A closed arrowhead ($\triangle$) indicates a typical surviving neuron, and an open arrowhead ($\blacktriangle$), a dead neuron. (B) Ratio of number of surviving neurons on ischemic side to number on contralateral (healthy) side. $^*P < 0.05$, $^{**}P < 0.01$. S, saline, $H_2$, hydrogen-rich saline; $\times 3$, three doses; $\times 7$, seven doses. Number of animals in each group: $H_2$-rich saline, n = 7 (three doses, n = 4; seven doses, n = 3); saline, n = 8 (three doses, n = 5; seven doses, n = 3).

and were stained darkly with hematoxylin. The ratio of hippocampal neurons on the ischemic side to the healthy contralateral side after three and seven doses was $0.058 \pm 0.013$ (n = 5) and $0.033 \pm 0.013$ (n = 3) in the saline group and $0.117 \pm 0.040$ (n = 4) and $0.149 \pm 0.071$ (n = 3) in the $H_2$-rich saline group. In seven dose groups, significantly more hippocampal neurons survived in the $H_2$-rich saline group than in the saline group.

## 3.2 Number of surviving neurons in cerebral cortex after ischemia-reperfusion

Ischemia-reperfusion injury markedly reduced the number of surviving neurons in layer III and V of the ischemic cerebral cortex compared with the contralateral healthy region (**Fig 2A and 2B**). The ratio of surviving neurons on the ischemic to the contralateral side in layer III of the cortex after three and seven doses was $0.371 \pm 0.153$ (n = 5) and $0.443 \pm 0.009$ (n = 3) in the saline group and $0.592 \pm 0.037$ (n = 4) and $0.687 \pm 0.024$ (n = 3) in the $H_2$-rich saline group (**Fig 2C**). The ratio of surviving neurons in the ischemic compared to the contralateral side in layer V of the cortex after 3 and 7 doses was $0.346 \pm 0.087$ (n = 5) and $0.404 \pm 0.040$ (n = 3) in the saline group and $0.585 \pm 0.117$ (n = 4) and $0.644 \pm 0.072$ (n = 3) in the $H_2$-rich saline group (**Fig 2D**). No difference was seen between the three- and seven-dose $H_2$-rich saline groups regarding the effectiveness of intraperitoneal $H_2$-rich saline injection in preventing neuronal death in the cerebral cortex.

## 3.3 Inhibition of neuronal apoptosis after ischemia-reperfusion

TUNEL-positive neurons were observed in the hippocampus on day 3 after ischemia-reperfusion (**Fig 3A**). We found no differences in the number of positive cells between the groups (**Fig 3B**). The positive neurons were presented in layer III and V of the ischemic cerebral cortex on day 3 after ischemia-reperfusion (**Fig 4A and 4B**). Significantly fewer TUNEL-positive cells in layer III were seen in the $H_2$-rich saline group (n = 4) than in the saline group (n = 3), but the number of TUNEL-positive cells in layer V was not significantly different between the two groups (**Fig 4C**).

## 3.4 $H_2$ concentration in the supra-hepatic IVC and carotid artery

The $H_2$ concentrations were measured in blood samples from the carotid artery and IVC 5 and 10 minutes after intraperitoneal administration of saline or $H_2$-rich saline (Fig 5, S1 Table). When saline was administered intraperitoneally, $H_2$ was detected in the IVC (mean, 0.00024 mg/L [0.0226%]) but not in the carotid artery in 5 of the 6 samples.

When saline was administered intraperitoneally, $H_2$ was detected in the IVC (mean, 0.00024 mg/L [0.0226%]) but not in the carotid artery in 5 of the 6 samples. After intraperitoneal administration of $H_2$-rich saline, $H_2$ concentrations increased significantly in the IVC (range, 0.00183 mg/L [0. 114%] to 0.00725 mg/L [0.453%]). In contrast, $H_2$ concentrations in the carotid artery remained in the range of 0.00008 mg/L (0.0049%) to 0.00023 (0.0146%). On average, $H_2$ concentrations in carotid artery were 0.04 times lower than in IVC.

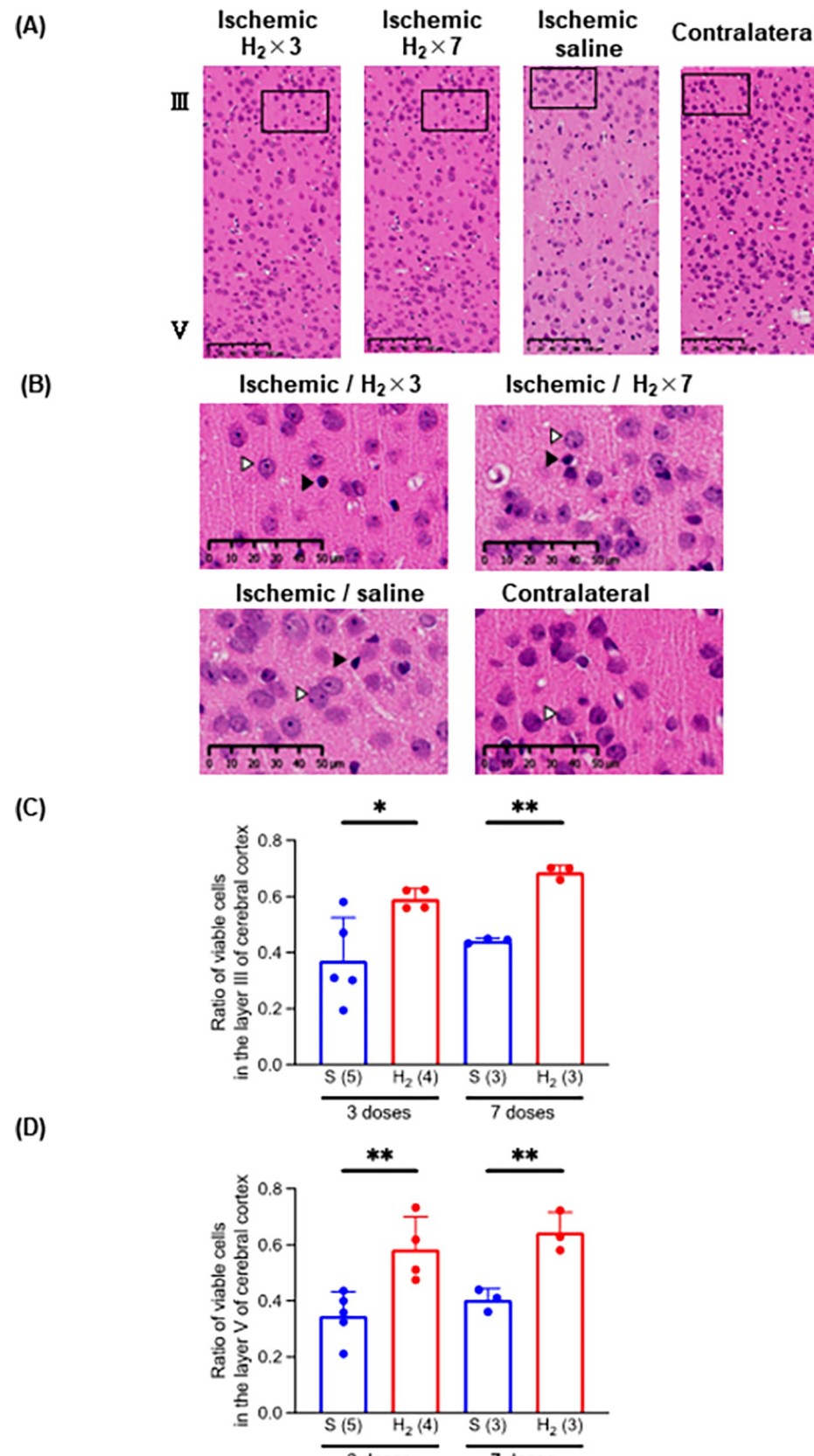

**Fig 2. Effect of intraperitoneal injection of H₂-rich saline on ischemic injury in gerbil cerebral cortex.** (A) Images of ischemic and contralateral (healthy) cortex stained with hematoxylin and eosin. (B) Higher magnification image of the inset in the image of the ischemic cortex and contralateral (healthy) cortex in (A). A closed arrowhead (△) indicates a typical surviving neuron, and an open arrowhead (▲), a dead neuron. (C) Ratio of number of surviving neurons in layer III of the cortex on ischemic side to number on contralateral (healthy) side. (D) Ratio of number of surviving neurons in layer V of the cortex on ischemic side to number on contralateral (healthy) side. $^{*}$ $P < 0.05$, $^{**}$ $P < 0.01$. S, saline, H₂, hydrogen-rich saline; × 3, three doses; × 7, seven doses. Number of animals in each group: H₂-rich saline, n = 7 (three doses, n = 4; seven doses, n = 3); and saline, n = 8 (three doses, n = 5; seven doses, n = 3).

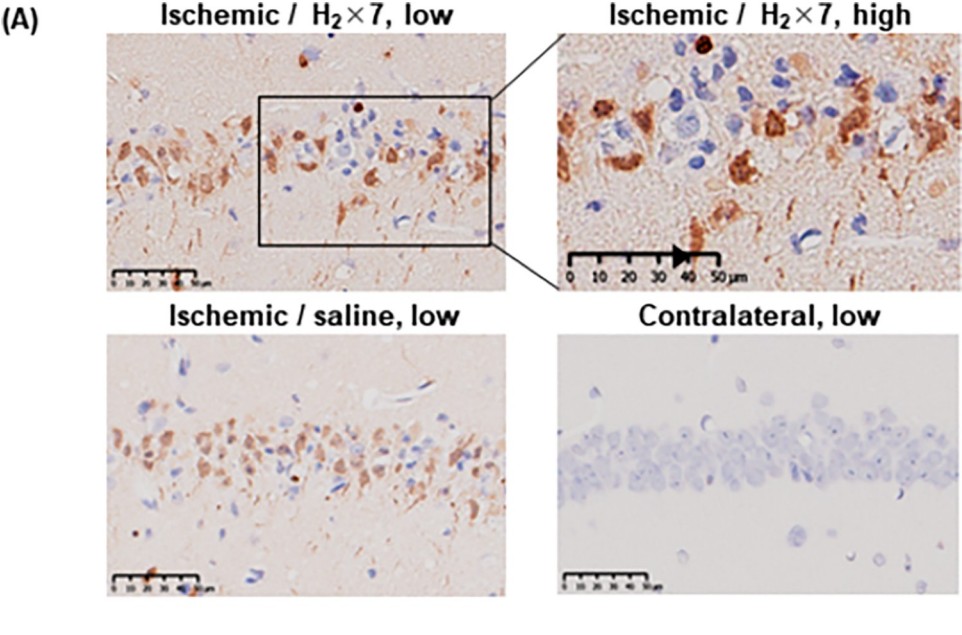

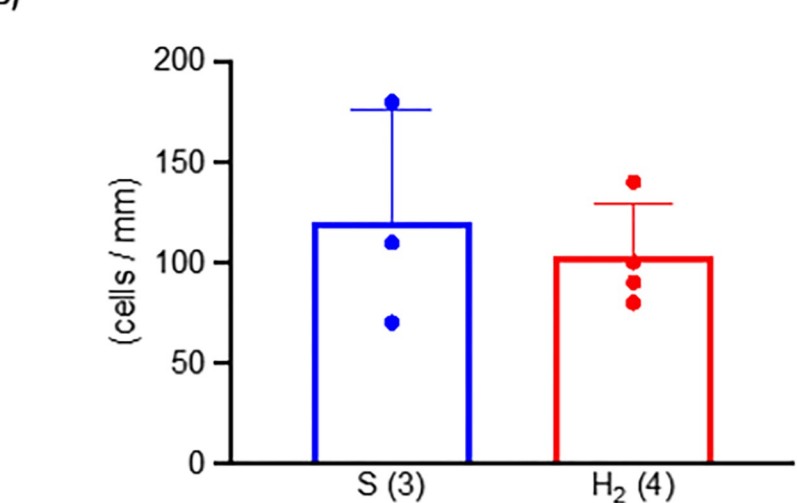

**Fig 3. Effect of intraperitoneal injection of H₂-rich saline on ischemic neuronal apoptosis in gerbil hippocampus.** (A) Images of ischemic and (healthy) contralateral hippocampus stained with terminal deoxynucleotidyl transferase dUTP nick end labeling (TUNEL). An arrowhead (▲) indicates a typical apoptosis cell (upper right). (B) Number of TUNEL-positive cells in ischemic hippocampus.

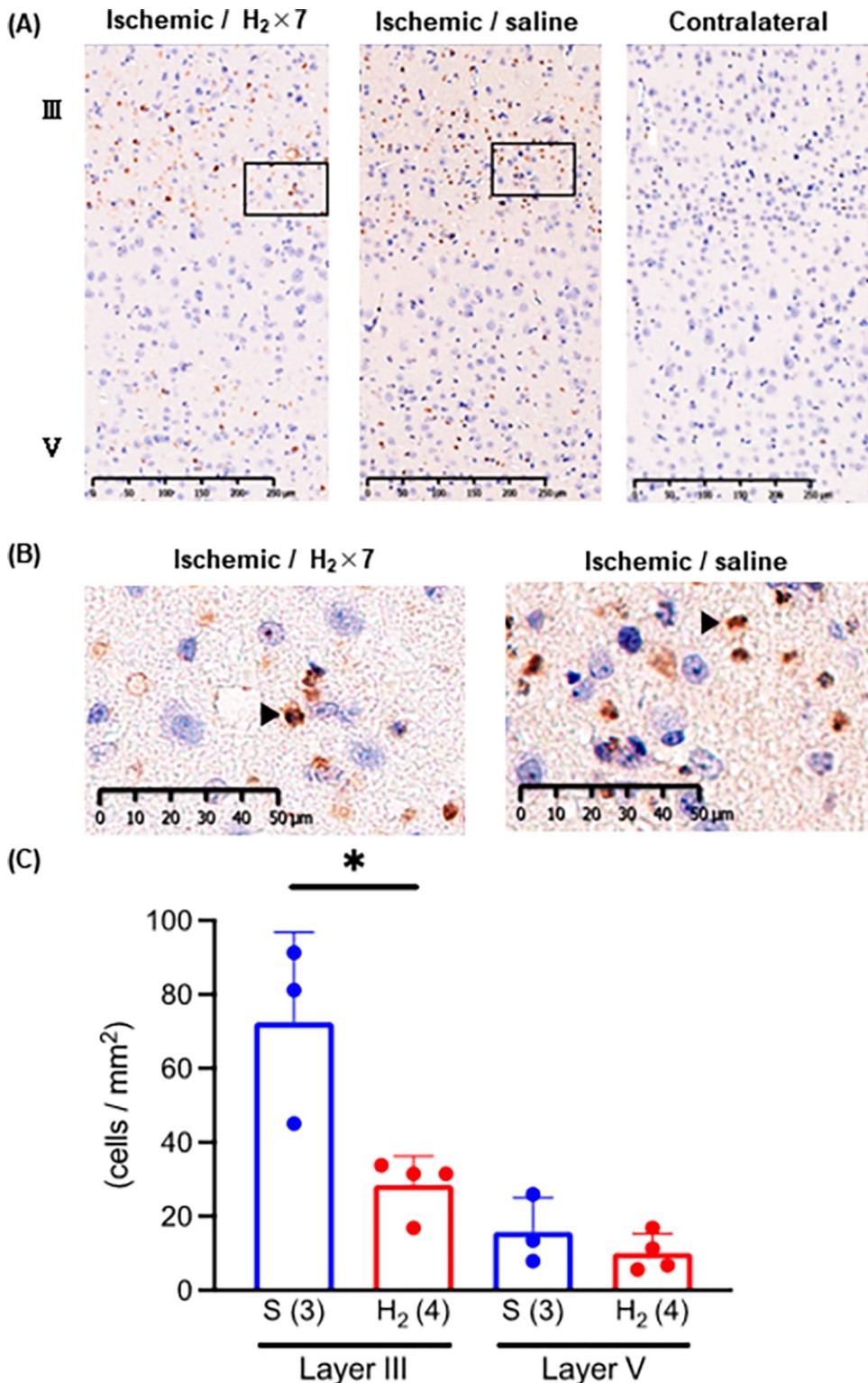

**Fig 4. Effect of intraperitoneal injection of H$_2$-rich saline on ischemic neuronal apoptosis in gerbil cerebral cortex.** (A) Images of ischemic contralateral and (healthy) cortex stained with terminal deoxynucleotidyl transferase dUTP nick end labeling (TUNEL). (B) Higher magnification image of the inset in the image of the ischemic cortex in (A). A typical apoptosis cell is indicated by an arrowhead (▲). (C) Number of TUNEL-positive cells in layers III and V of the cerebral cortex. * $P < 0.05$. S, saline, H$_2$, hydrogen-rich saline; × 7, seven doses. Number of animals in each group: H$_2$-rich saline, n = 4; saline, n = 3.

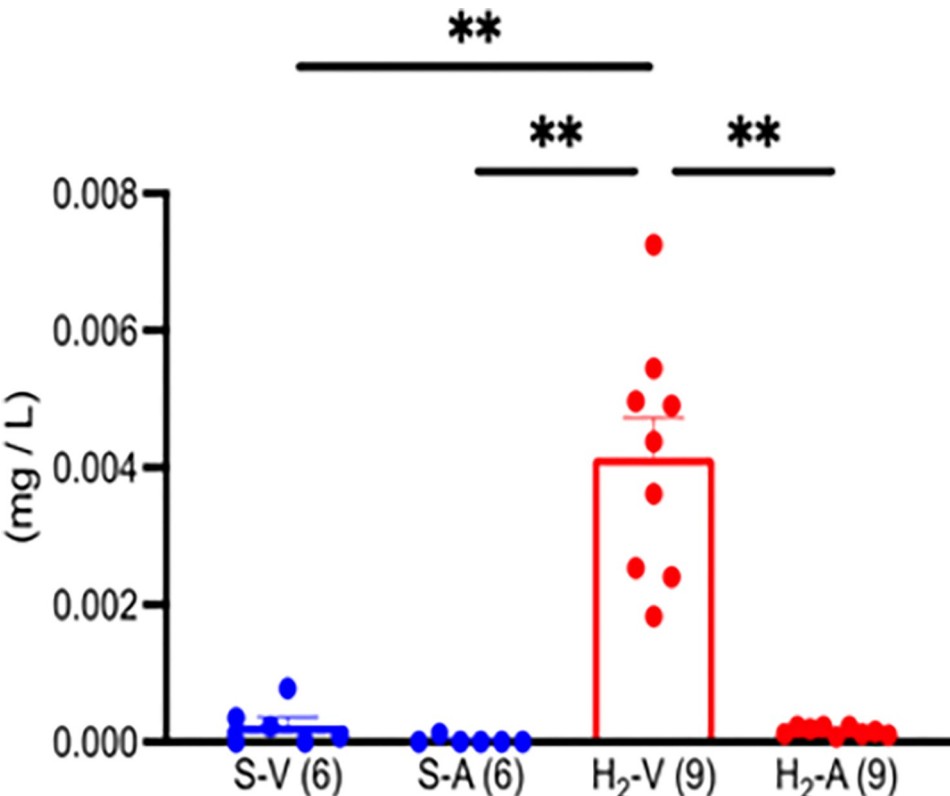

**Fig 5. $H_2$ concentration in the inferior vena cava and carotid artery of gerbils after intraperitoneal administration of $H_2$-rich saline and saline.** The not detectable $H_2$ concentration was the same as that of the blank and was calculated as zero in the statistical test. $**$ $P < 0.01$. A, carotid artery; $H_2$, hydrogen-rich saline; S, saline; V, inferior vena cava; Number of animals in each group: $H_2$-rich group, n = 9; saline group, n = 6.

## Discussion

In the present study, we showed that repeated, brief cerebral ischemia reproducibly induces neuronal death in the ischemia-vulnerable hippocampus and layers III and V of the cerebral cortex and that intraperitoneal administration of $H_2$-rich saline inhibits neuronal cell death in these brain regions. The number of surviving hippocampal neurons after ischemic recanalization was significantly higher after seven than after three doses of $H_2$-rich saline, indicating a dose-response relationship.

Although several studies have examined the efficacy of intraperitoneal administration of $H_2$-rich saline on ischemia-reperfusion injury in the brain [18, 20–23], this study is novel in that it examined the effects on individual neuronal death in the hippocampus and cerebral cortex, rather than on infarct size reduction, in an experimental gerbil system, a highly reliable cerebral ischemia model [15, 16].

The second novelty of this study is that it determined the pharmacokinetics of $H_2$ when injected intraperitoneally as $H_2$-rich saline. Specifically, blood was injected into a sealed vial, and the vaporized hydrogen was measured by gas chromatography.

The $H_2$ concentration of each sample was expressed as a corrected value obtained after subtracting the $H_2$ concentration in air [4, 11, 12]. The lower detection limit of the analyzer was 0.1 ppm, but the $H_2$ value in air was 0.5 ppm, well above the quantitation limit. Therefore, each sample containing $H_2$ from air and blood should have a value greater than 0.5 ppm, i.e., above the quantitation limit. If the analyzer shows the hydrogen concentration in a sample as

0.5 ppm, that sample contains only $H_2$ from air and no $H_2$ from blood; in this case, the measured value would be described as not detectable (n.d.).

After intraperitoneal administration of saline, a mean of 0.00024 mg/L (0.0226%) of $H_2$ was detected in the IVC, but no $H_2$ was detected in the carotid artery. The $H_2$ in the IVC is thought to be derived from intestinal bacteria [24–28]. We assume that most of the $H_2$ was expelled with exhaled air while passing through the lungs and therefore did not reach the carotid artery.

Intraperitoneal administration of $H_2$-rich saline resulted in a large increase in $H_2$ concentration in the IVC (from 0.00183 mg/L [0.114%] to 0.00725 mg/L [0.453%]). We suggest that $H_2$ in intraperitoneally administered $H_2$-rich saline was absorbed by the portal venous system and subsequently reached the IVC. Experiments in pigs have shown that two thirds of the $H_2$ in the portal venous blood are absorbed in the liver and the remainder reaches the IVC [4]. From these results, it can be concluded that intraperitoneal administration of $H_2$-rich saline is a good model that can mimic the condition of drinking $H_2$-rich water.

Intraperitoneal administration of $H_2$-rich saline resulted in only very slight increases (ranging from 0.008 mg/L [0.0049%] to 0.00023 [0.0146%]) in $H_2$ concentrations in the carotid artery. Unlike in pigs, in gerbils frequent blood sampling is too invasive, so carotid artery $H_2$ concentrations were measured only once, 10 minutes after intraperitoneal administration of $H_2$-rich saline. Previously, when an $H_2$-rich solution was administered to the jejunum of minipigs, we observed that the $H_2$ concentration in the portal blood increased after 2 minutes and continued to rise to approximately the same level at 5, 10, 20, and 30 minutes. Therefore, it is unlikely that the timing of the peak carotid $H_2$ concentration was missed.

$H_2$ may have been absorbed from the injected $H_2$-rich saline into the intraperitoneal blood and transported via the portal vein, liver, and IFC to the lungs, where it was excreted almost entirely during exhalation. These results are similar to those of our previous study in which hydrogen water was injected directly into the jejunal lumen of pigs [4].

In both clinical trials and animal studies, inhalation of $H_2$ is regulated to achieve a hydrogen concentration of 1% to 4% in the inhaled air [29]. Currently, it is unclear whether the inhibitory effect of $H_2$ on ischemic neuronal cell death observed in this study is explained by a minute amount of $H_2$, about 0.01 times the amount inhaled, reaching the brain or by an indirect effect via an inter-organ communication [30]. Further studies are needed.

## Supporting information

**S1 Fig. Anatomical comparison of basilar artery in rat and gerbil.** In rats (left), the internal carotid artery system, vertebral artery system, and left and right medial artery systems are connected at the base of the brain to form the circle of Willis. Therefore, if one artery is occluded, blood can still flow to the brain via the other vessels. In the gerbil (right), the basilar arterial ring lacks anastomoses between the right and left internal carotid arteries and between the carotid and vertebral arteries, so occlusion of the common carotid artery can easily induce ipsilateral cerebral ischemia.
(TIF)

**S2 Fig. Experimental protocols.** (A) Experimental protocol-1. After two 10-minute occlusions of the left carotid artery, the animals were randomly divided into H2-rich saline and saline groups. Animals in each group received 3 or 7 intraperitoneal injections of 20 ml/kg body weight of each solution. Histological analysis was performed on days 3 and 7. (B) Experimental protocol-2. H2-rich saline or saline solution (20 ml/kg BW) was administered intraperitoneally once. Blood was drawn from the carotid artery after 5 minutes and from the IVC after 10 minutes, and H2 concentration was measured by gas chromatography.
(TIF)

**S3 Fig. Measurement site of the number of survived neurons in the brain of gerbils after transient cerebral ischemia.** Cerebral histology of the cerebrum of a gerbil coronally sectioned at the levels of the infundibulum and stained with hematoxylin and eosin. Neuronal cells were counted in the hippocampal CA1 region (areas inside the blue rectangles) and the central part of the cerebral cortex (areas inside the black rectangles, 300 μm wide) in the left and right cerebral hemispheres.
(TIF)

**S1 Table. H2 concentration in the inferior vena cava and carotid artery of gerbils after intraperitoneal administration of H2-rich saline.** H2 measurements are shown in concentration (mg/L) and saturation (%). Because H2 dissolves at 1.6 mg/L under 0.1 MPa, the H2 saturation of each sample was converted by assuming 1.6 mg/L as 100% saturation. IVC, inferior vena cava, CA carotid artery, n.d., no detected, i.p., intraperitoneal injection.
(TIF)

## Acknowledgments

The authors thank Sou Hashimoto and Yasuyo Aoyama (Doctors Man Co., Ltd.), Shigeo Ohta (Japanese Molecular Hydrogen Promotion Association [JHyPA]), Suga Kato (JHyPA), and Mayumi Takeda (JHyPA) for technical assistance.

## Author Contributions

**Conceptualization:** Yoji Hakamata, Motoaki Sano.

**Data curation:** Momoko Hirano, Yoji Hakamata.

**Formal analysis:** Momoko Hirano.

**Funding acquisition:** Momoko Hirano, Yoji Hakamata.

**Investigation:** Momoko Hirano, Yoji Hakamata.

**Methodology:** Momoko Hirano, Kazuhisa Sugai, Yoji Hakamata.

**Project administration:** Yoji Hakamata, Motoaki Sano.

**Resources:** Yoji Hakamata, Motoaki Sano.

**Supervision:** Masahiko Fujisawa, Eiji Kobayashi, Yoshinori Katsumata.

**Validation:** Momoko Hirano, Kazuhisa Sugai, Eiji Kobayashi, Yoshinori Katsumata, Yoji Hakamata, Motoaki Sano.

**Visualization:** Momoko Hirano, Kazuhisa Sugai, Eiji Kobayashi, Yoshinori Katsumata, Yoji Hakamata, Motoaki Sano.

**Writing – original draft:** Yoji Hakamata, Motoaki Sano.

**Writing – review & editing:** Momoko Hirano, Kazuhisa Sugai, Eiji Kobayashi, Yoshinori Katsumata, Yoji Hakamata, Motoaki Sano.

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
