## [Decision Letter · Decision Letter 0]

7 Sep 2022

PONE-D-22-22387Hydrogen indirectly inhibits ischemic neuronal cell death in the gerbil brain after intraperitoneal administration of hydrogen-rich salinePLOS ONE

Dear Dr. Sano,

Thank you for submitting your manuscript to PLOS ONE. Your study has been evaluated by two experts in the field. After careful consideration, we feel that it has merit but does not fully meet PLOS ONE’s publication criteria as it currently stands. Therefore, we invite you to submit a revised version of the manuscript that addresses the points raised during the review process.

The major concerns of two reviewers and requests of Academic Editor that fall in line with the PLOS ONE publication expectations are outlined below:

[1] (Academic Editor) Your publication does not adhere to PLOS ONE Data Sharing Policy, as your primary results have not been included in supplemental files or made available through open database depository (such as OSF, Figshare, or Dryad).  See additional requirements at https://journals.plos.org/plosone/s/data-availability.  If your resubmission does not meet this criterion (provides access to primary information included in all statistical analyses) , it will be rejected outright.

[2] One of the major points of your submission is not fully supported by experimental data.  Specifically, I refer to close-to-undetectable levels of hydrogen in carotid circulation (and by extension in the brain) and the speculation that such levels are too low to explain protective effects of systemic hydrogen delivery.  Both reviewers questioned reliability of this conclusion and sensitivity of hydrogen detection method.  Mors specifically:

- (Reviewer 2) Your baseline hydrogen measurement was done only once.  This is insufficient for making conclusions and statistical comparisons.   Please add additional baseline measurements and run statistics comparisons.  

- (Reviewer 1) Sensitivity of the method may not be sufficient to detect increases in the intracarotid hydrogen levels.  Please add additional discussion in the text.

- (Academic Editor)  Due to short half-life of hydrogen in arterial circulation (~90 seconds), it is not clear if the selected timing for the intraarterial hydrogen measurements truly reflect transients of this gas upon intraperitoneal delivery in gerbils.  Please additionally discuss this in the text.

[3] (Academic Editor) In IHC assays, do you discriminate reliably between pycnotic and healthy nuclei?  This distinction is not obvious in provided representative images.  In Fig. 1 and Fig 2, please include additional insets with high magnification allowing for better discrimination between healthy and dying cells and include markings indicating healthy and injured/dead cells.[4] (Reviewer 1) TUNEL staining quantification was performed in context but not vulnerable hippocampus.  If the relevant IHC slides/scans are preserved, please add hippocampal quantification or explain why it is impossible.

[5] (Reviewer 1, Review 2, Academic Editor) Your speculation that hydrogen provides indirect protection in the brain by acting on parasympathetic and sympathetic innervation is solely a speculation, which has not been tested by the presented experiments.  For this reason, please exclude references to indirect mechanisms and gut-brain interactions from abstract, discussion, and concluding statements in the manuscript.

- For example, on lines 358-360 you write [quotation] “Although we were able to show that intraperitoneally administered H2-rich saline protects brain neurons indirectly via the gut-brain connection, i.e., an inter-organ network…”  This is an example of unacceptable statement that is not supported by experimental evidence in the manuscript.</li></ul>

We look forward to receiving your revised manuscript.

Kind regards,

Alexander A. Mongin, Ph.D.

Academic Editor

PLOS ONE

2. We note that you have a patent relating to material pertinent to this article. Please provide an amended statement of Competing Interests to declare this patent (with details including name and number), along with any other relevant declarations relating to employment, consultancy, patents, products in development or modified products etc. Please confirm that this does not alter your adherence to all PLOS ONE policies on sharing data and materials, as detailed online in our guide for authors http://journals.plos.org/plosone/s/competing-interests by including the following statement: "This does not alter our adherence to PLOS ONE policies on sharing data and materials.” If there are restrictions on sharing of data and/or materials, please state these. Please note that we cannot proceed with consideration of your article until this information has been declared.

Reviewers' comments:

Reviewer's Responses to Questions

**Comments to the Author**

1. Is the manuscript technically sound, and do the data support the conclusions?

Reviewer #1: Partly

Reviewer #2: Partly

2. Has the statistical analysis been performed appropriately and rigorously? 

Reviewer #1: Yes

Reviewer #2: No

3. Have the authors made all data underlying the findings in their manuscript fully available?

Reviewer #1: No

Reviewer #2: Yes

4. Is the manuscript presented in an intelligible fashion and written in standard English?

Reviewer #1: Yes

Reviewer #2: Yes

5. Review Comments to the Author

Reviewer #1: In this manuscript, using intraperitoneal injection of hydrogen saline, HE staining and Tunel staining, the authors found that hydrogen saline injection can reduce ischemia caused by many times in hippocampus and cortex neuron loss, reduce the amount of cortex cell apoptosis, determination of portal vein and hydrogen content of the internal carotid artery, portal vein hydrogen content increased, However, hydrogen in carotid artery had no detected. Based on these results, the authors concluded that intraperitoneal injection of hydrogen saline may have a protective effect on brain injury caused by cerebral ischemia through indirect effects, but the specific mechanism is not confirmed.

 This is interesting work, However there are some problems should be considered.

According to the results of brain histological staining, the author verified the protective effect of intraperitoneal injection of hydrogen saline on cerebral ischemia, but the evidence was relatively thin. Although there were similar studies on the protection of hydrogen water on cerebral ischemia in the past, it should be supported by more research basis to be more convincing. For example, behavioral studies to verify changes in protein levels of apoptosis.

As for hydrogen acting through indirect effects, the conclusion obtained in this study is not rigorous enough. 1. There are differences in the effect of different doses and times of administration, and the dose-effect relationship is more supportive of direct effect, but certainly cannot be used as evidence to negate indirect effect. 2. The authors could not put only hydrogen from the internal carotid artery as a basis, and believed that hydrogen did not act directly on the brain, but guessed that it was realized through indirect effects such as the brain-gut axis.

Hydrogen could not be measured, possibly because the sensitivity of the method was not high enough. The author can measure in the portal vein to hydrogen, hydrogen can be absorbed in the abdominal cavity and enter the blood circulation, considering that the hydrogen is insufficient evidence of animal cell metabolism, hydrogen in animals is the inertia of the mantra, hydrogen is released only when pulmonary circulation, but such release cannot be no residual, less is retained, Resulting in undetectable detection by insensitive assays. If very little hydrogen enters the brain still can produce protective effect against cerebral ischemia injury, in other words, small dose and large effect can indicate that the effect of hydrogen is very strong, of course, indirect effect cannot be ruled out. If an indirect effect is to be judged, research evidence of such an effect should be provided, rather than simply relying on insufficient and rigorous evidence to guess.

A previous study by the authors[ref] found that inhaled hydrogen could measure higher levels in the internal carotid artery. Intraperitoneal injection of hydrogen saline could not measure the amount of hydrogen in the internal carotid artery. Given that the same group found that hydrogen inhalation could measure sufficient amounts of hydrogen in the internal carotid artery, inhalation should be used as a control to observe differences in the protective effects of hydrogen entry into the brain against neuronal loss following ischemic injury. However, the manuscript does not make such a comparative study.

Tunel staining only has cortical data, not hippocampal results. Why is this? This study has cortex and hippocampus

HE staining, but Tunel staining only in the cortex, there are no staining in hippocampus, which is not readily understood.

M. Sano, G. Ichihara, Y. Katsumata, T. Hiraide, A. Hirai, M. Momoi, T. Tamura, S. Ohata, E. Kobayashi

Pharmacokinetics of a single inhalation of hydrogen gas in pigs

PLoS One, 15 (2020), Article e0234626

Ichihara G, Katsumata Y, Moriyama H, Kitakata H, Hirai A, Momoi M, Ko S, Shinya Y, Kinouchi K, Kobayashi E, Sano M. Pharmacokinetics of hydrogen after ingesting a hydrogen-rich solution: A study in pigs. Heliyon. 2021 Nov 11;7(11):e08359. doi: 10.1016/j.heliyon.2021.e08359. PMID: 34816046; PMCID: PMC8591508.

Reviewer #2: Methodologically, this is a solid manuscript with appropriate controls and methods used except for the H2 concentration data. The manuscript is logical and well written.

Major concerns:

1. The study is confirmative. The reported data just confirm previously published data but on a different rodent stroke model and no new scientific information is reported. The manuscript would benefit from addressing the mechanism for H2 protective effect.

2. Table 1 data (H2 concentration): n=1 for saline group is not sufficient; statistical analysis is not performed.

Minor suggestions:

1. Blood H2 concentration assay should be described in more details

6. PLOS authors have the option to publish the peer review history of their article (what does this mean?). If published, this will include your full peer review and any attached files.

Reviewer #1: No

Reviewer #2: No

---

## [Author Response · Author response to Decision Letter 0]

10 Nov 2022

Point-by-point response to the academic editor’s and reviewers’ comments

Thank you for your constructive comments and suggestions, which have helped to improve our manuscript. Our point-by-point responses are shown below:

Responses to the Academic Editor and Reviewers 

#1 Academic Editor

Your publication does not adhere to PLOS ONE Data Sharing Policy, as your primary results have not been included in supplemental files or made available through open database depository

Author Response: In accordance with the PLOS ONE Data Sharing Policy, we have revised all bar graphs as bar graph + error bars + dot blots to clearly show individual data (Figure 1 to Figure 4). The results of the hydrogen concentration measurements are shown in Figure 5 as bar graph + error bars + dot blots and were statistically analyzed. These raw data are presented in the supplementary table.

#2 Reviewer 2 comment

Your baseline hydrogen measurement was done only once. This is insufficient for making conclusions and statistical comparisons. Please add additional baseline measurements and run statistics comparisons.

Author Response: To clarify the changes in blood hydrogen concentration after intraperitoneal administration of H2-rich saline, 3 new animals were added to the hydrogen group (total: 9 animals) and 5 animals to the saline group (total: 6 animals). The n.d. (not detectable) point was calculated as zero for statistical tests. The raw data of the hydrogen concentration measurements are shown in the supplementary table, and to clarify the experimental results, we created a new Figure 5 showing the results of the statistical tests.

#2 Reviewer 1 comment

Sensitivity of the method may not be sufficient to detect increases in the intracarotid hydrogen levels. Please add additional discussion in the text.

Author Response: To measure blood H2 concentration, we first inserted a needle into the rubber lid of a 13.5-mL sealed vial, extracted 1 mL of air, and injected 1 mL of blood. To prevent outgassing, we immediately applied wax to the rubber lid to seal the injection hole. In such a sealed vial, H2 in the blood is released into the air phase because the air in the vial contains almost no H2 gas, so after injection of blood, most of the H2 gas moves from the blood (liquid phase) into the air (gas phase). Therefore, examining the concentration of H2 gas in the air inside the vial enabled the H2 concentration in the blood to be estimated. Some of the air phase (0.2 mL, 0.4 mL, or 1 mL, depending on the H2 concentration) was collected from the vial, and the H2 concentration was measured by gas chromatography (TRIlyzer mBA-3000, Taiyo, Co., Ltd.). A calibration curve was obtained by using H2 gas at a concentration of 0 (nitrogen gas), 5, 50, and 130 parts per million (ppm). The detectable value of the analyzer is 0.1 ppm (i.e., the analyzer shows values of 0.1 ppm or higher), and the limit of quantitation is considered to be approximately 0.2 ppm. 

Samples were collected from the carotid artery after 5 minutes and from the inferior vena cava after 10 minutes, and each sample was measured twice. At the same time, air was collected from the exact same location and injected into a blood-free vial, and the H2 gas concentration was measured. The value of H2 in the air was 0.5 ppm, i.e., above the limit of quantitation. The air value was subtracted from each measured sample value to obtain a corrected value. Thus, only samples with an H2 value above 0.5 ppm contained H2 also from blood. 

In the revised manuscript, the Methods section describes in detail the method used for measuring hydrogen concentration. In the Discussion section, we also explain the sensitivity of the H2 concentration measurement.

#2 Academic Editor comment

Due to short half-life of hydrogen in arterial circulation (~90 seconds), it is not clear if the selected timing for the intraarterial hydrogen measurements truly reflect transients of this gas upon intraperitoneal delivery in gerbils. Please additionally discuss this in the text.

Author Response: Thank you for pointing out this important point. In a previous study, when we administered an H2-rich solution to the jejunum of minipigs, we observed that the H2 concentration in the portal blood was increased at 2 minutes and continued to rise to 2 minutes and then remained at approximately the same level at 5, 10, 20, and 30 minutes. Therefore, it is unlikely that the timing of the peak carotid H2 concentration was missed. In the revised manuscript, we now explain the above in Discussion section to justify the timing of the H2 concentration measurement.

#3 Academic Editor comment

In Fig. 1 and Fig 2, please include additional insets with high magnification allowing for better discrimination between healthy and dying cells and include markings indicating healthy and injured/dead cells.

Author Response: Thank you for this suggestion. We have added a high magnification image in Figure 1-4 and have marked healthy and dying cells in it so that the reader can distinguish between them.

#4 Reviewer 1 comment

TUNEL staining quantification was performed in context but not vulnerable hippocampus. If the relevant IHC slides/scans are preserved, please add hippocampal quantification or explain why it is impossible.

Author Response: We have followed the Reviewer's instructions and added the results of TUNEL staining of the hippocampus as Figure 3. The number of TUNEL-positive cells in the hippocampus was measured and compared between the two groups. At day 3 of ischemia-reperfusion injury, there was no significant decrease in the number of TUNEL-positive cells after intraperitoneal administration of H2-rich saline.

#5 Reviewer 1, Review 2, Academic Editor comments

Please exclude references to indirect mechanisms and gut-brain interactions from abstract, discussion, and concluding statements in the manuscript. 

Author Response: As directed by Reviewer 1, Reviewer 2, and the Academic Editor, we have excluded statements on indirect mechanisms and gut-brain interactions, as well as the cited references. We have completely rewritten the Discussion.

#5 Reviewer 1 comment

If an indirect effect is to be judged, research evidence of such an effect should be provided, rather than simply relying on insufficient and rigorous evidence to guess.

Author Response: We completely agree with this comment. As directed by Reviewer 1, Reviewer 2, and the Academic Editor, we have excluded statements on indirect mechanisms and gut-brain interactions, as well as the cited references. Intraperitoneal administration has been the route of administration most frequently used for observing the effects on rodents of H2 in H2-rich saline. To the best of our knowledge, no studies have evaluated the pharmacokinetics of H2 administered by this route. We hope that our study will help to elucidate the mechanism of action of H2.

HE staining, but TUNEL staining only in the cortex, there are no staining in hippocampus, which is not readily understood.

Author Response: We have added the results of TUNEL staining of the hippocampus as Figure 3. The number of TUNEL-positive cells in the hippocampus was measured and compared between the two groups. At day 3 of ischemia-reperfusion injury, there was no significant decrease in the number of TUNEL-positive cells after intraperitoneal administration of H2-rich saline.

#5 Reviewer 2 comment

The study is confirmative. The reported data just confirm previously published data but on a different rodent stroke model and no new scientific information is reported. The manuscript would benefit from addressing the mechanism for H2 protective effect.

Author Response: Intraperitoneal administration has been widely used as a route of administration for observing the brain-protective effects in rodents of H2 in H2-rich saline. To our knowledge, this article is the first to report on the pharmacokinetics of H2 administered by this route of administration. In addition, this study is novel in that it examined the effects of intraperitoneal administration of H2-rich saline on individual neuronal death in the hippocampus and cerebral cortex—rather than reduction in the size of necrotic brain tissue—in a gerbil experimental system, which is a reliable model of cerebral ischemia.

Table 1 data (H2 concentration): n=1 for saline group is not sufficient; statistical analysis is not performed.

Author Response: We completely agree with this comment. To clarify the changes in blood hydrogen concentration after intraperitoneal administration of H2-rich saline, we added 3 new animals to the H2 group (total: 9 animals) and 5 to the saline group (total: 6 animals). The n.d. (not detectable) point was calculated as zero for statistical tests. The raw data of the hydrogen concentration measurements are shown in the supplementary table, and to clarify the experimental results, we created a new Figure 5 showing the results of the statistical tests.

Minor suggestions:

Blood H2 concentration assay should be described in more details.

Author Response: Thank you for your important comment. To measure blood H2 concentration, we first inserted a needle into the rubber lid of a 13.5-mL sealed vial, extracted 1 mL of air, and injected 1 mL of blood. To prevent outgassing, we immediately applied wax to the rubber lid to seal the injection hole. In such a sealed vial, H2 in the blood is released into the air phase because the air in the vial contains almost no H2 gas, so after injection of blood, most of the H2 gas moves from the blood (liquid phase) into the air (gas phase). Therefore, examining the concentration of H2 gas in the air inside the vial enabled the H2 concentration in the blood to be estimated. Some of the air phase (0.2 mL, 0.4 mL, or 1 mL, depending on the H2 concentration) was collected from the vial, and the H2 concentration was measured by gas chromatography (TRIlyzer mBA-3000, Taiyo, Co., Ltd.). A calibration curve was obtained by using H2 gas at a concentration of 0 (nitrogen gas), 5, 50, and 130 parts per million (ppm). The detectable value of the analyzer is 0.1 ppm (i.e., the analyzer shows values of 0.1 ppm or higher), and the limit of quantitation is considered to be approximately 0.2 ppm.

Samples were collected from the carotid artery after 5 minutes and from the inferior vena cava after 10 minutes, and each sample was measured twice. At the same time, air was collected from the exact same location and injected into a blood-free vial, and the H2 gas concentration was measured. The value of H2 in the air was 0.5 ppm, i.e., above the limit of quantitation. The air value was subtracted from each measured value to obtain a corrected value. Thus, only samples with a value above 0.5 ppm contained H2 also from blood. 

In the revised manuscript, the Methods section describes in detail the method of measuring hydrogen concentration. In the Discussion section, we also explain the sensitivity of the H2 concentration measurement.

---

## [Decision Letter · Decision Letter 1]

7 Dec 2022

Pharmacokinetics of hydrogen administered intraperitoneally as hydrogen-rich saline and its effect on ischemic neuronal cell death in the brain in gerbils

PONE-D-22-22387R1

Dear Dr. Sano,

We’re pleased to inform you that your manuscript has been judged scientifically suitable for publication and will be formally accepted for publication once it meets all outstanding technical requirements.

Kind regards,

Alexander A. Mongin, Ph.D.

Academic Editor

PLOS ONE

Additional Editor Comments (optional):

Reviewers' comments:

Reviewer's Responses to Questions

**Comments to the Author**

1. If the authors have adequately addressed your comments raised in a previous round of review and you feel that this manuscript is now acceptable for publication, you may indicate that here to bypass the “Comments to the Author” section, enter your conflict of interest statement in the “Confidential to Editor” section, and submit your "Accept" recommendation.

Reviewer #1: All comments have been addressed

Reviewer #2: All comments have been addressed

2. Is the manuscript technically sound, and do the data support the conclusions?

Reviewer #1: Partly

Reviewer #2: Yes

3. Has the statistical analysis been performed appropriately and rigorously? 

Reviewer #1: No

Reviewer #2: Yes

4. Have the authors made all data underlying the findings in their manuscript fully available?

Reviewer #1: Yes

Reviewer #2: Yes

5. Is the manuscript presented in an intelligible fashion and written in standard English?

Reviewer #1: Yes

Reviewer #2: Yes

6. Review Comments to the Author

Reviewer #1: The author responded to the review comments very seriously, and there have been many improvements in the new manuscript, but there are still some problems that have not been clarified, and even new problems have emerged in the process of revision, requiring the author to give explanations.

The basic conclusion of this study is that adding hydrogen to the abdominal pathway can have a protective effect on brain injury, but the changes in hydrogen concentration detected by carotid artery cannot explain the protective effect of hydrogen entering the brain. This conclusion is very difficult to understand. In order to reach such a conclusion, that is, to rule out the direct effect of hydrogen, in addition to the problem of detecting the sensitivity of the hydrogen method, there should also be evidence of this indirect effect, or that hydrogen is likely to play an indirect role in the abdominal cavity through an indirect route. There are many possibilities for indirect effects. For example, it may affect the immune inflammatory response through phytonerves. It also works through the blood circulation by influencing the release of other biomolecules from the gut flora. However, it is better that these effects should be studied accordingly, otherwise there is still a possibility that the conclusion cannot be understood.

After hydrogen is ingested by the body, there are three metabolic directions: one is captured and fixed by local tissues; the other is transferred with blood circulation. In this study, hydrogen ingested through the abdomen will mainly regress to the heart with blood and then enter the pulmonary circulation, most of which may spread into the air. There is also some hydrogen circulating from the skin to the outside world. Third, it is metabolically consumed by body tissues. All three cases have been reported in academic studies. But either way, you can't completely metabolize hydrogen. Any metabolic pattern in the body probably follows the rule of inverse exponential decay, which is that as the concentration decreases, the metabolic efficiency decreases. In other words, hydrogen is unlikely to disappear from the body in a very short time. However, with any detection method, there is a problem of detection sensitivity. In this study, hydrogen could not be analyzed in the internal carotid artery, mainly due to the detection sensitivity. This study adopts gas chromatography, which is a classical hydrogen analysis method. But the sensitivity of this analysis is also very limited. For example, the concentration of air can only be analyzed above 0.5ppm. While the headspace method itself has an effect on the efficiency of releasing hydrogen from the liquid, which in this study is blood. This may be the reason why this study could not detect sufficient levels of hydrogen, and also the reason why the changes could not be analyzed.

Regarding the peak hydrogen concentration, the authors report that "when an H2-rich solution was applied to the jejunum of the mini pigs, we observed that the portal blood H2 concentration increased at 2 minutes, continued to rise to 2 minutes, and then remained at roughly the same level at 5,10,20, and 30 minutes." The total body volume of an animal is a significant factor affecting the gas diffusion. The gas conversion patterns of gerbils and pigs are quite different, and the results measured in pigs should not be used to invert the results in mice. Just as you can't classify people by the gas diffusion patterns in an elephant's body.

For the problem of insufficient sample size in the original experiment, the author revised the feedback saying that the sample size was increased, so as to meet the requirements of statistical analysis. However, this completely violates the basic principle of experimental design. Statistical analysis requires that samples should not be randomly added in the middle, because the principle of random grouping cannot be realized in this way. If the author must do so, he must make this modification clear in the manuscript.

Reviewer #2: (No Response)

7. PLOS authors have the option to publish the peer review history of their article (what does this mean?). If published, this will include your full peer review and any attached files.

Reviewer #1: No

Reviewer #2: No

---

## [Editor Report · Acceptance letter]

15 Dec 2022

PONE-D-22-22387R1 

Pharmacokinetics of hydrogen administered intraperitoneally as hydrogen-rich saline and its effect on ischemic neuronal cell death in the brain in gerbils 

Dear Dr. Sano:

I'm pleased to inform you that your manuscript has been deemed suitable for publication in PLOS ONE. Congratulations! Your manuscript is now with our production department. 

Kind regards, 

on behalf of

Dr. Alexander A. Mongin 

Academic Editor

PLOS ONE